# Leveraging publicly available coronavirus data to identify new therapeutic targets for COVID-19

**Stacy L. Sell**[1], **Donald S. Prough**[1], **Harris A. Weisz**[1], **Steve G. Widen**[2], **Helen L. Hellmich**[1]*

**1** Department of Anesthesiology, The University of Texas Medical Branch at Galveston, Galveston, Texas, United States of America, **2** Department of Biochemistry and Molecular Biology, The University of Texas Medical Branch at Galveston, Galveston, Texas, United States of America

* hhellmic@utmb.edu

## Abstract

Many important questions remain regarding severe acute respiratory syndrome coronavirus-2 (SARS-CoV-2), the viral pathogen responsible for COVID-19. These questions include the mechanisms explaining the high percentage of asymptomatic but highly infectious individuals, the wide variability in disease susceptibility, and the mechanisms of long-lasting debilitating effects. Bioinformatic analysis of four coronavirus datasets representing previous outbreaks (SARS-CoV-1 and MERS-CoV), as well as SARS-CoV-2, revealed evidence of diverse host factors that appear to be coopted to facilitate virus-induced suppression of interferon-induced innate immunity, promotion of viral replication and subversion and/or evasion of antiviral immune surveillance. These host factors merit further study given their postulated roles in COVID-19-induced loss of smell and brain, heart, vascular, lung, liver, and gut dysfunction.

## Introduction

In the midst of the current coronavirus disease-19 (COVID-19) pandemic [1], three baffling questions remain. *First*, why are so many COVID-19-positive individuals asymptomatic [2, 3]? The answer is critical because asymptomatic carriers and symptomatic patients shed similar amounts of virus [4]. *Second*, what underlying genetic mechanisms account for individual variability in disease susceptibility and mortality? Male sex and old age were among the greatest risk factors associated with COVID-19, but the full scope of risk factors is unknown [5]. *Third*, why do as many as 74% of recovered COVID-19 patients suffer months of chronic debilitating symptoms such as dyspnea and fatigue [6], reflecting long-term effects of SARS-CoV-2 infection on virtually every organ system [7]?. We hypothesize that COVID-19-positive patients with chronic symptoms are deficient in multiple antiviral host factors. Given the urgency of this pandemic and the knowledge that similar sequelae have been observed in the 2002 SARS and 2012 MERS outbreaks [8], we propose to capture a broader understanding of pro-viral and anti-viral host response factors via *in silico* analysis of publicly available SARS and MERS

**Data Availability Statement:** All data are in the publicly available Gene Expression Omnibus database, in the paper and in the supplemental information.

**Funding:** The authors received no specific funding for this work.

**Competing interests:** The authors have declared that no competing interests exist.

datasets in the Gene Expression Omnibus (GEO) and expand the range of potential treatment options. Since there are many peer-reviewed studies that have extensively compared the clinical and pathological similarities and differences among SARS, MERS and COVID-19 (for example, see [9]), the focus of this present analysis of existing coronavirus datasets in GEO is on identifying potentially novel molecular targets via examination of genes and miRNA target genes that are affected by these different coronaviruses in infected host cells.

We used principal component analysis (PCA) [10], a powerful data reduction statistical method, combined with hierarchical clustering of the discriminating variables that represent the top three principal components, for bioinformatic analysis of four coronavirus datasets in the GEO that may provide both common [9] and unique insights into pathogenic mechanisms. We recently used these methods to show that blood microRNA (miRNA) expression profiles can identify potential blood markers of a broad spectrum of human diseases [11]. We examined four GEO datasets that represent two previous deadly coronavirus outbreaks, Middle East Respiratory Syndrome coronavirus (MERS-CoV) [GSE81852], Severe Acute Respiratory coronavirus 1 (SARS-CoV-1) [GSE1739], a comparison of SARS-CoV-1 with Dhori virus, a member of the orthomyxovirus family that includes the influenza viruses [GSE17400] and lastly, a comparison of SARS-CoV-1 with SARS-CoV-2 [GSE148729]. Two of these datasets show direct mRNA (gene) expression changes induced by viral infections. The two miRNA datasets were analyzed for two reasons: 1) the limited number of discriminating miRNAs may prove to be both potential biomarkers of coronavirus infections as well as potential targets of antiviral therapeutics and 2) the predicted gene targets of these miRNA variables would provide valuable insight into dysregulated pro-viral and antiviral host responses that could also be therapeutically targeted. Following PCA and hierarchical clustering, we performed literature mining for all discriminating gene variables and for all 1506 predicted gene targets of five differentially expressed miRNAs that distinguish MERS or SARS infected samples from healthy controls. We found evidence that a diverse range of host factors are co-opted or deregulated by coronaviruses to 1) promote viral replication, assembly and production in host cells and 2) to subvert or evade the host antiviral innate and adaptive immune responses. Importantly, we identify genes whose dysregulation may be consistent with SARS-CoV-2-induced heart [12], lung/vascular [13], brain/cognitive [14], liver [15] and gut [16] dysfunction and notably, genes linked to loss of smell in acute and chronic COVID-19 patients.

## Materials and methods

### Overview

Unsupervised statistical methods such as principal component analysis (PCA) [10] and hierarchical clustering [17] are used to study data structure and look for patterns, i.e. clusters of samples. Coronavirus gene (mRNA) and miRNA datasets were downloaded from the National Center for Biotechnology Information's Gene Expression Omnibus database and PCA and hierarchical clustering were done on these datasets using Qlucore Omics Explorer (Qlucore AB, Lund Sweden), an interactive bioinformatic data analysis program which combines powerful statistical analysis (based on R) with instant visualization for 2D or 3D presentations of data in real time.

**Variance filtering, PCA, hierarchical clustering and miRNA target prediction.** For each dataset, prior to PCA and hierarchical clustering analyses, variance filtering was used to reduce the noise, and the projection score was used to set the filtering threshold. The "Mean = 0, Var = 1" setting has been used to scale the data. N variables were left after the filtering. The projection score [18] was used to determine the optimal filtering threshold, retaining N variables.

Principal Component Analysis (PCA) [10] was used to visualize the data in a 3D space, after filtering out variables with low overall variance to reduce the impact of noise, and centering and scaling the remaining variables to zero mean and unit variance. Results generated with selected p value cutoffs and q-values (false discovery rate) [19] were also visualized by hierarchical clustering (using average linkage method [20]) which was used to group samples and (or) variables by similarity in an unsupervised manner.

To identify predicted miRNA gene targets whose functional roles can be analyzed by literature mining, lists of differentially expressed miRNAs were inputted into a target prediction algorithm, miRDB [21]. Through a proprietary bioinformatics program, miRDB uses a machine-learning based approach to assess functional annotations for miRNAs and their respective gene targets. A prediction score ranging from 50–100 is then assigned to the miRNA–gene relationship with higher scores indicating more confidence in the predicted relationship. Gene targets were then filtered based on prediction score (>60) and curated lists of genes were studied by literature mining.

## Results and discussion

We first examined a MERS-CoV dataset of miRNA expression in primary human airway epithelial cells infected with wild-type MERS. Differential expression of only two miRNAs (miR-4697-5p, downregulated and miR-4521, upregulated by coronavirus infection, respectively) was sufficient to distinguish all MERS-infected cells from mock-infected cells (Fig 1). For mechanistic insights into viral pathogenesis, we compiled a list of gene targets of these two miRNAs using miRDB, an online miRNA target prediction database that derives functional annotations based on computational and literature mining [21]. We then studied the biological relevance to viral pathogenesis and long-term sequelae based on published literature. The miRNA target genes, their functional roles and supporting literature citations are described in S1 Table; in this narrative, numbers in parentheses following gene symbols (full gene names provided in S1 Table) correspond to the numbered genes in S1 Table. The three key findings are 1) the majority of the miRNA gene targets are known to promote viral pathogenesis but some have antiviral functions, 2) some of these genes targets are druggable with existing drugs, and 3) the functional roles of these host response factors (viral replication, subversion of antiviral signaling, immune cell evasion, injury to major organ systems such as the brain and heart) are recurring themes in all four coronavirus datasets.

Among the pro-viral host genes, targeted by miR-4697-5p, that are predicted to be upregulated in infected host cells, are those implicated in promoting transcription and replication of hepatitis B virus (HBV), CRTC1 (#9), hepatitis C virus (HCV), SLC12A4 (#21) and influenza A virus (IAV), NDRG1 (#22); negative regulators of antiviral responses, i.e. USP21 (#36) that suppresses antiviral interferon signaling; regulators of clotting, THPO (#8), consistent with reports of abnormal clotting in COVID-19 patients [22]. Given reports of neurological and cognitive impairment as a result of COVID-19 [14], we identified several genes implicated in inflammatory neuronal injury, L1CAM (#15), neurodegenerative disorders, DLG4 (#11), Zika-induced microcephaly, MSI1 (#18), and neurocognitive disorders, ATP1A3 (#24); genes implicated in cardiomyopathy, HMGA1 (#17), lung epithelial injury response, FOXP4 (#4) and inflammatory lung injury, PALM (#27); a regulator of T cell development and antiviral immune responses, EFNB1 (#2), an instigator of T cell exhaustion during chronic viral infections, IL2RB (#34), and a regulator of immunoglobulin secretion from B cells, CPLX2 (#10); and among the protective host genes are those involved in interferon-inducible innate antiviral immune responses, TRIM25 (#7). Given the existence of drug inhibitors of MMPs [23], one of the potentially druggable targets is MMP15 (#12); recent reports showed that MMP-induced

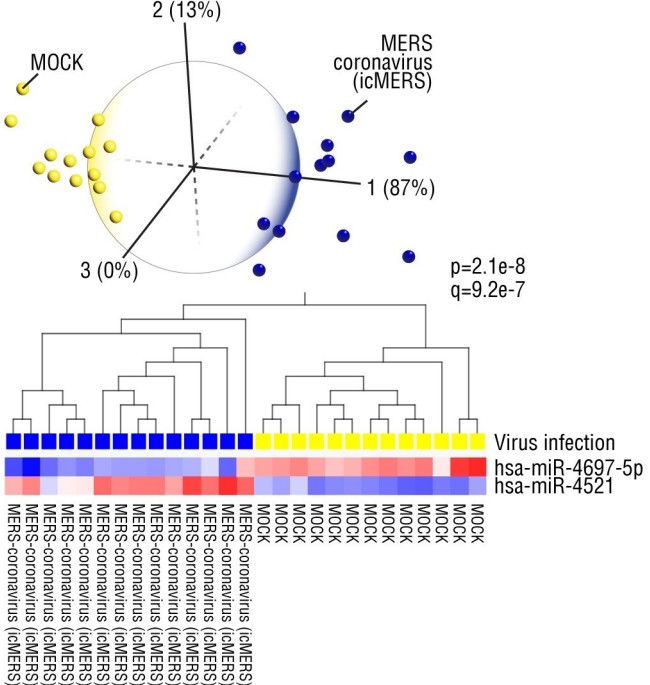

**Fig 1. MERS-CoV-induced microRNA (miRNA) expression in primary human airway epithelial cells (GSE81852).**
Two differentially expressed miRNAs (hsa-miR-4697-5p, hsa-miR-4521) clearly distinguish MERS-coronavirus
(icMERS) infected from MOCK infected control cells; over 50% of predicted gene targets of both miRNAs are
implicated in pro-viral, antiviral and/or immunomodulatory functions. Expression levels analyzed using principal
component and hierarchical clustering analyses.

destruction of the extracellular matrix may increase risk of myocardial infarction in COVID-19 patients with underlying cardiovascular disease [24, 25]. In all, given the upregulation of miR-4697-5p in all healthy controls and the ease of detecting elevated levels of biomarkers versus decreased levels, we suggest that high levels of miR-4697-5p, found in all uninfected cells, could serve as a potential diagnostic marker of coronavirus disease resistance in the asymptomatic population.

As we would find in the other coronavirus datasets, miRNA target gene expression was remarkably consistent with disease identification; increased expression of miR-4521 in MERS-infected cells would be predicted to suppress expression of antiviral target genes, such as HIPK2 (#522), KLF5 (#545) and FBXO21 (#608). Among the great diversity of miR-4521 gene targets, we found multiple genes known to regulate organ systems impacted by SARS-CoV-2; genes implicated in diabetes, KLF6 (#517), FBXO28 (#551) and TTC39A (#542); implicated in lung inflammation, WWOX (#557); implicated in cardiomyopathy, ENO1 (#547); and multiple genes linked to congenital heart disease and heart failure, DCAF5 (#572), CLIC2 (#577), TOR1AIP1 (#618), PRKD1 (#629). Notably, given reports that loss of smell is a prominent and long-lasting symptom in COVID-19 patients [26], it is striking that loss-of-function of TENM1 (#521) and OLFM1 (#587) is known to cause congenital anosmia and defective olfaction, respectively.

We next performed PCA and hierarchical clustering of mRNA (gene) expression data from peripheral blood mononuclear cells (PBMCs) of patients and healthy controls from the 2002–2003 SARS outbreak (Fig 2). As we found in the MERS dataset, the genes that discriminated the SARS patients from controls are associated with pro-viral or antiviral functions and some

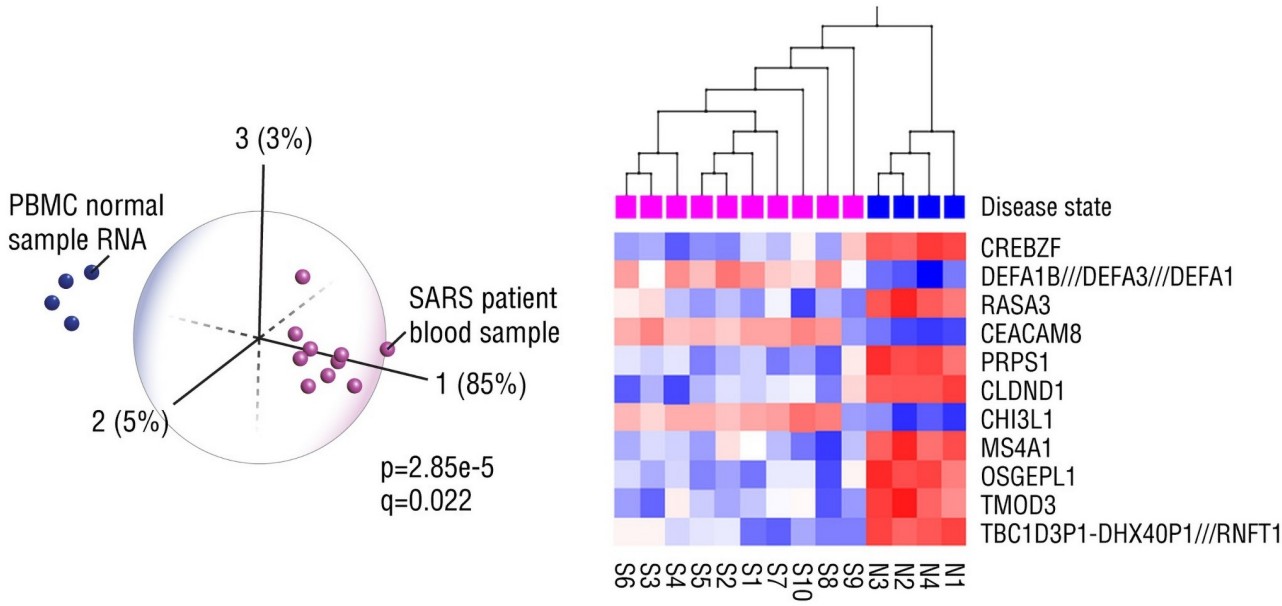

**Fig 2. Gene (mRNA) expression in Peripheral Blood Mononuclear Cells (PBMC) obtained from SARS-CoV-1-infected patient blood samples (GSE1749).** Eleven genes, involved in innate immune responses and antiviral defense, clearly distinguish normal (N1-4) from SARS patient blood samples (S1-10). Expression levels analyzed using principal component and hierarchical clustering analyses.

differentially expressed genes have been shown to have similar roles in the pathogenesis of several diverse viruses such as herpes simplex virus (HSV) and human immunodeficiency virus (HIV) as well as coronavirus. For instance, the transcriptional regulator, CREBZF, that inhibits HSV DNA replication and suppresses expression of HSV immediate-early and late genes [27], was expressed at high levels in all the healthy controls. CREBZF also regulates the unfolded protein response (UPR) [28] that is important for innate immunity and viral sensing as well as the pathogenesis of a broad range of human diseases [29] and can be therapeutically targeted because its expression can be induced by glucose or insulin [30]. Another druggable target is the antimicrobial and antiviral peptide, defensin-1 alpha (DEFA1), that is known to support the innate immune response by disrupting viral membranes [31]. The upregulation of DEFA1 in SARS-infected cells is a protective antiviral host response that could be augmented with existing drugs and natural compounds that induce DEFA1 expression [32]. Other antimicrobial peptides such as lactoferrin, a component of breast milk, have shown antiviral activity against SARS-CoV [33] and it is noteworthy that a recent analysis of SARS-CoV-2 in breast milk from 18 infected women did not detect replication-competent virus in any of the samples [34]. Given the increased risk of thrombosis and coagulation disorders in COVID-19 patients [35], the high expression of RASA3, an inhibitor of platelet activation, in healthy PBMCs may be clinically significant if also found in infected but asymptomatic individuals. Interestingly, CEACAM8 that is highly upregulated in SARS patients and downregulated in all healthy controls was previously shown to be a potential biomarker of HBV vaccine non-responders [36] and may be relevant to the clinical evaluation of future SARS-CoV-2 vaccines. We identified a SARS-upregulated gene, CHI3L1, that, if also upregulated in COVID-19 patients, could be responsible for COVID-19-induced complications [6]. This gene is already strongly associated with, and plays a major role in, tissue injury, inflammation, tissue repair and remodeling responses in diseases such as asthma, arthritis, sepsis, diabetes, liver fibrosis, and coronary artery disease [37].

We next examined mRNA expression data from a study of 2B4 cells, a derivative of human bronchial epithelial cells (Calu-3), that compared the dynamic host innate immune response to SARS-CoV-1 and Dhori virus (DHOV) infections (Fig 3). DHOV is a member of the Ortho-myxoviridae family of RNA viruses that includes all four genera that cause influenza in humans, birds, and mammals. As in the MERS and SARS datasets, the discriminating genes that identified the SARS-infected and/or Dhori-infected cells can be broadly linked to pro- or anti-viral functions. However, the key insight into coronavirus pathogenesis came from the temporal gene expression profile induced by these viruses. Twelve hours after infection, FOS,

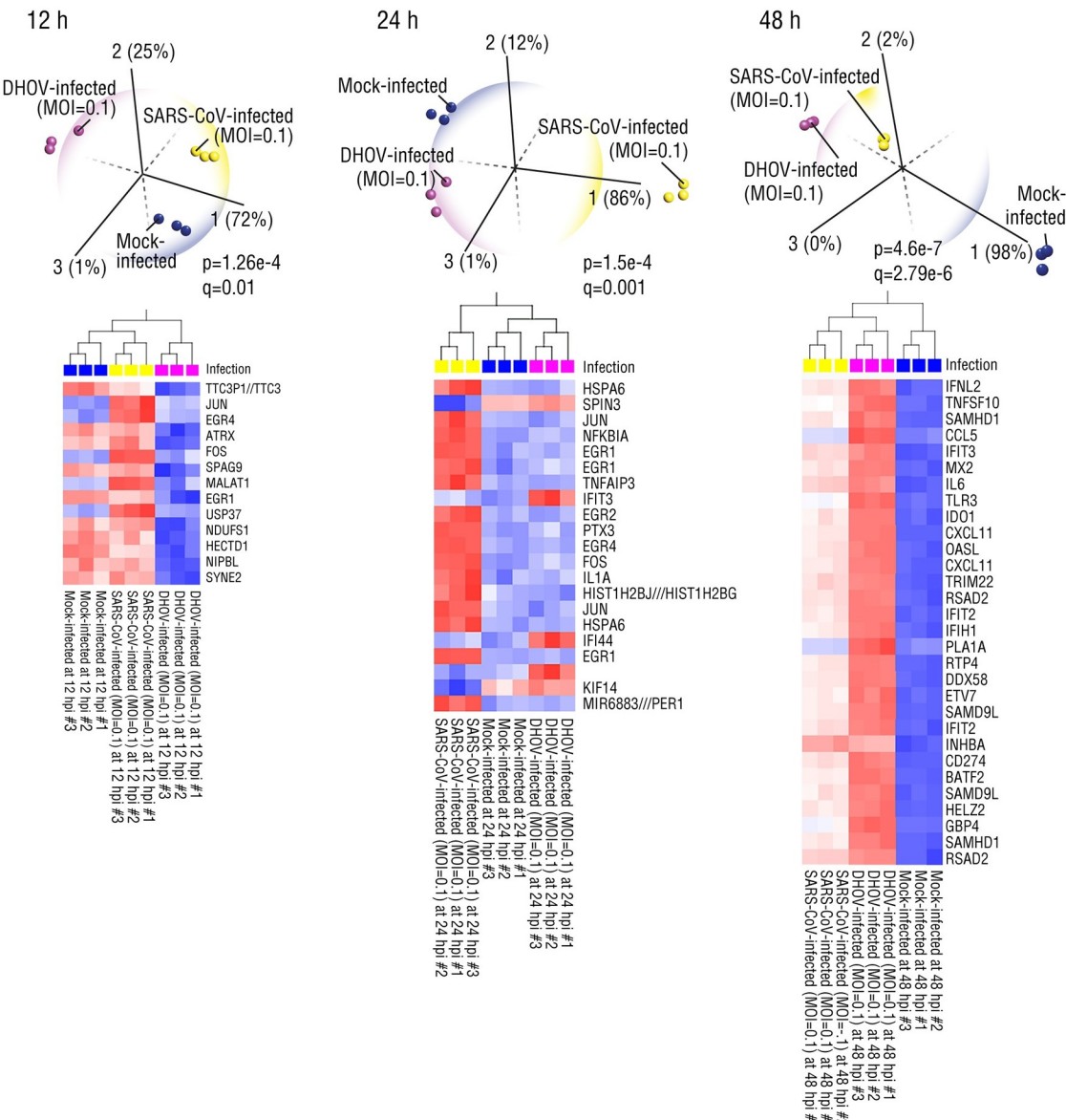

**Fig 3. Dynamic temporal (12. 24, 48 hour) gene expression profiles distinguish coronavirus from Dhori virus-infected human bronchial epithelial (2B4) cells (GSE17400).** Principal component and hierarchical clustering analyses of SARS-CoV-infected, Dhori virus- (DHOV) infected and Mock-infected Calu-3 cells shows that at each time point, the dynamic gene expression profiles, 1) discriminates virus-infected from Mock-infected cells, 2) distinguishes between the two virus infected groups from each other, and 3) identify discriminating genes that are involved in transcriptional regulation of the host immune response. Hpi = hours post infection, MOI = multiplicity of infection.

JUN, EGR1, and EGR4, transcriptional regulators that are activated in response to pathogens were activated in SARS but not DHOV infected cells, suggesting a rapid host response was elicited by coronavirus infection. Among the many mechanisms that viruses have evolved to evade host antiviral responses are those targeting interferons that are essential for host defense against viral infection. Indeed, dysregulation of interferon expression underlies many autoinflammatory diseases including viral infections [38]; at 24 hours post infection, two interferon-induced antiviral genes, IFI44 and IFIT3, were significantly suppressed in SARS but not DHOV-infected cells. By 48 hours, we found that although the transcriptional profile of upregulated genes was similar in SARS and DHOV-infected cells, expression of antiviral genes such as IFNL2 was attenuated only in SARS-infected cells. Moreover, two genes, the inflammatory cytokine, CCL5 and the phospholipase, PLA1A that are normally induced by viral infection were downregulated in SARS but not DHOV-infected cells. Because CCL5 is involved in activation of the innate immune response [39] and knockdown of PLA1A expression suppresses the innate immune signaling induced by RNA viruses [40], this suggests that SARS-CoV-1 rapidly inhibits the innate antiviral immune response. One possible mechanism of this inhibition was suggested by a study showing that SARS-CoV-1 appears to manipulate host mitochondrial function to evade host innate immune responses [41]; mitochondrial signaling has been shown to control innate and adaptive immune responses [42]. Thus, the severity of SARS coronavirus infections may be linked to the rapid suppression of host innate immunity, suggesting that timely augmentation of the innate immune response could have therapeutic benefits [43]. Recently, an article in Nature by Finkel et al., showed clear evidence that SARS-CoV-2 impairs the translation of multiple host transcripts, including innate immune genes, that are induced in response to viral infection [44].

A recent study of factors that explained individual host responses to SARS-CoV-2 showed that men and elderly people are more likely to die from COVID-19 than women because they have delayed immune responses [45]. Notably, the immune systems of women <60 years old produced a rapid and near immediate defense against SARS-CoV-2; in contrast, in men of all ages, immune cell responses were not activated until after three days following the onset of infection [45]. Moreover, immune cell composition and function fluctuated with viral loads in males and the elderly, i.e., a marked reduction in cytotoxic T and natural killer cells, indicative of a dysfunctional antiviral response. Together with reports that women develop higher levels of coronavirus-fighting T cells [46], these two studies are congruent with our finding that unlike Dhori virus, SARS-CoV-infected host cells can be distinguished by the rapid suppression of intracellular viral detection mechanisms that would induce interferon production. We infer that this rapid temporal suppression allows viral replication to occur in the first two days of infection before the host immune system is fully mobilized to fight the virus [45].

Analysis of the fourth coronavirus dataset, miRNA expression in human airway epithelial (Calu-3) cells infected with SARS-CoV-1 or SARS-CoV-2 (Fig 4), confirmed that coronaviruses could exploit diverse host mechanisms for replication and survival. A diverse array of host genes that are targets of miR-139-3p, miR-301a-5p and miR-1290 have been hijacked to facilitate immune evasion and suppression (S1 Table). Furthermore, analysis of gene targets of miR-139-3p, that is the only miRNA expressed at low levels (thus, the target genes would be predicted to be upregulated) in both SARS-CoV-1 and SARS-CoV-2 cells as well as untreated control cells, led us to speculate that the normal, upregulated expression levels of some host genes could provide a permissive environment for viral infections, i.e., viruses could exploit and coopt the endogenously expressed host proteins to facilitate entry into host cells and favor the replication of viral proteins: ARFGEF2 (#37) is linked to poliovirus replication, VAPB (#39) is involved in HCV and norovirus replication, ATL1 (#53) promotes human immunodeficiency virus (HIV) and Zika virus infection, and TBC1D20 (#55) is involved in replication

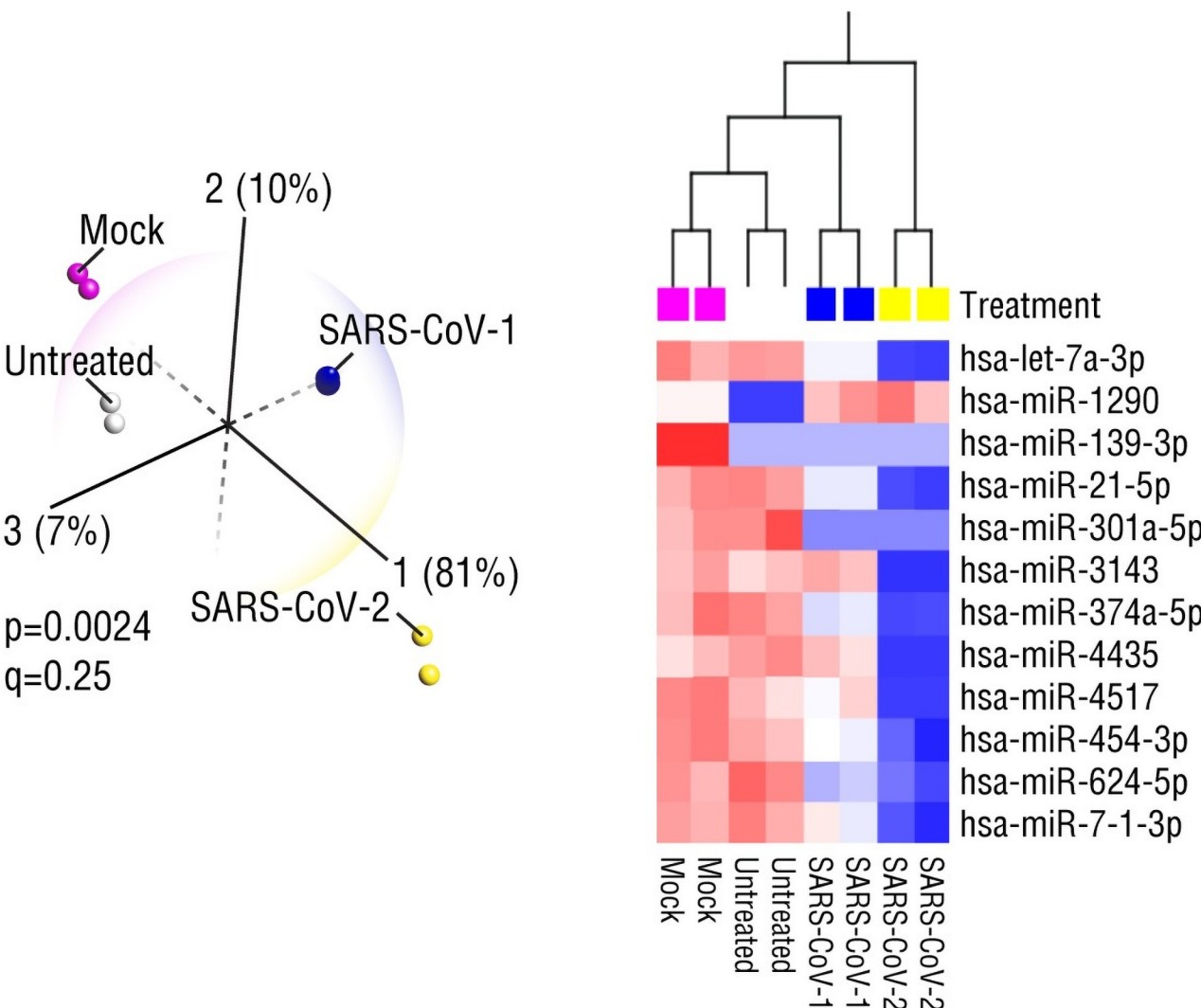

**Fig 4. MicroRNA (miRNA) expression profiles identify similarities and differences between SARS-CoV-1 and SARS-CoV-2 infections of human airway epithelial cells (GSE148729).** Four hours after infection with SARS-CoV-1, SARS-CoV-2, Mock virus or untreated, principal component and hierarchical clustering analyses showed that the four groups can be distinguished from one another based on differential expression of twelve miRNAs. Nine miRNAs distinguish SARS-CoV-1 from SARS-CoV-2, while three miRNAs are similarly expressed in SARS-CoV-1 and SARS-CoV-2 samples. Literature mining of all predicted miRNA target genes showed that over 50% have demonstrated functional roles in the host response to viral infection.

and assembly of HIV, HSV1 and HCV in host cells, SIRPA (#42) is a negative regulator of innate immune response, i.e., downregulates type 1 interferons and UVRAG (#41) is needed for entry of Ebola and other filoviruses into host cells. However, expression of several antiviral factors, such as PACSIN1 (#52) that suppresses HCV infection, PRDM9 (#54), that is involved in innate antiviral immunity, and RELT (#56), that activates NF-κB antiviral immune signaling, indicate that the functional effect of miR-139-3p would depend on the ratio of pro- to anti-viral gene targets. As in the MERS study, multiple target genes could contribute to viral pathogenesis: ARHGEF28 (#51) and CD6 (#48) are known to regulate T cell and B cell function, DCLK1 (#40) is implicated in HCV replication and virus-induced liver inflammation, SDK1 (#45) is linked to HIV-induced neuropathy and SLC2A2 (#50) is implicated in virus-induced diabetes.

Likewise, we find that multiple predicted gene targets of miR-301a-5p - downregulated by both SARS viruses so gene targets are predicted to be upregulated—are hijacked by diverse viruses to promote viral replication: of Chikungunya and HIV, DHX9 (#304), of Aichi virus (positive-strand RNA picornavirus), PITPNB (#343), of HIV, CCNT2 (#338), and replication and assembly of IAV and HSV1, RAB1A (#342); and to promote viral transport of Japanese encephalitis virus, SGMS1 (#336), and infection by HCV, ERLIN1 (#344). Interestingly, because downregulation of ZEB1 (#306), that is induced by SARS-CoV-2 in lung cancer cells, is associated with protection against idiopathic pulmonary fibrosis, the predicted upregulation of this gene suggests that viral infection can worsen lung function. TICAM2 (#387) that has a role in type 1 interferon signaling, has also been shown to contribute to SARS-CoV pathogenesis. Among the genes that play an antiviral/protective function are HMGN2 (#358) that inhibits HBV expression and replication, TNPO1 (#346) that is required for interferon-mediated antiviral functions, i.e. HIV restriction, and NFYB (#305) that restricts replication of human T cell lymphotropic virus-2 (HTLV-2) in host cells. As found with MERS, multiple disease-associated genes are implicated in virus-induced damage to different organs; i.e. coronary artery disease, ZNF347 (#368); myocardial infarction and liver disease, BMP2 (#331); autoimmune neurological disorders, PNMA1 (#308); chronic lung inflammatory disease, CCM2 (#337) and TOM1L1 (#352), and a marker of lung infection with respiratory syncytial virus, ZNF322 (#327); XPNPEP3 (#351) whose increased expression is linked to neuropsychiatric disorders, RBFOX1 (#389) whose increased expression could impair memory and cognitive function and ANKS1B (#315), an Alzheimer's gene that is also a host integration site for HBV. Given reports of sudden-onset diabetes in COVID-19 patients [47], we identified B4GALT5 (#362) that is increased in diabetes and obesity and HIF1A (#365) that has a role in virus-induced metabolic changes that lead to type 2 diabetes.

Although miR-1290, the only miRNA significantly upregulated in both virus-treated groups, was previously shown to be upregulated by influenza virus and enhance viral replication and polymerase activity [48], the functional roles of its gene targets suggest that it has both antiviral and pro-viral functions; of the likely inhibited targets are pro-viral host factors that promote infection by adenovirus, KLC1 (#72), by flavivirus, CBLL1 (#67), by respiratory syncytial virus (RSV), MS4A3 (#83), by Seneca Valley virus, ANTXR1 (#107), by dengue virus, SCP2 (#125), by human cytomegalovirus (CMV), HHEX (#229) and by varicellar zoster virus, IDE (#190); some target genes are involved in replication of HIV, COG5 (#93) and RANBP1 (#89), involved in infection, replication and assembly of classical swine fever virus, RAB18 (#115), and involved in infection and transport of enterovirus and herpes simplex virus (HSV), PICALM (#117). Because many target genes, such as MAP3K2 (#139), function as hubs of inflammatory signaling, their inhibition could mitigate the systemic inflammatory response to SARS-CoV-2 infection. On the other hand, inhibition of endogenously protective host factors could be deleterious, i.e., numerous miR-1290-inhibited target genes are also known to have interferon-inducible antiviral activity against HIV, CD164 (#92), antiviral activity against and clearance of IAV, SOCS4 (#95) and downregulation of OTUD4 (#250) would decrease levels of type 1 interferons and proinflammatory cytokines and potentiate viral replication.

As observed in the MERS and SARS datasets, there are a striking number of potentially inhibited miR-1290 gene targets that regulate adaptive responses in a wide range of immune cells; CAMK4 (#97), a regulator of activated B cells ACKR4 (#111), three regulators of T cell signaling, TNIK (#113), RASGRF2 (#130), and SH2D1A (#134), a regulator of interaction between T cells and dendritic cells, TNFRSF11A (#226) and MALT1 (#127) that controls rabies virus infection by inducing inflammation and T cell activation. Multiple host gene targets are associated with protective or deleterious roles in different organs; ERBB4 (#90) is

protective in liver and heart, PRKAA2 (#180) is cardioprotective, FPGT (#146) is implicated in viral myocarditis, overexpression of CRNDE (#227) attenuates cardiac fibrosis and enhances cardiac function, PHLDA1 (#237) is implicated in oxidative stress-induced cardiomyocyte injury and myocardial ischemia reperfusion injury, FIGN (#235) protects against congenital heart disease, SPATA9 (#133) is implicated in COPD, dysregulation of PTK2 (#231) together with Sox11 plays a role in ventilator-induced lung injury, FOXA2 (#161) regulates liver function and inflammatory genes in muscle and other tissues, and CYTH2 (#189) has a role in inflammatory cell activation and recruitment and disruption of vascular stability. Among the central nervous system (CNS) targets are DCX (#122), a neurodevelopment gene that is down-regulated by Zika virus and linked to defects in brain development after Zika virus infection, MED23 (#157), an antiviral gene whose altered expression is a risk for cognitive decline and dementia, NLGN3 (#206) that is involved in synaptic plasticity and implicated in neurodevelopmental disorders such as autism and several Alzheimer's disease genes with roles in HIV, BACE1 (#207) and herpes virus infections, APPBP2 (#208). Among the druggable targets is the identification of negatively regulated genes that are already known to be targeted by antiviral drugs, e.g., BCL7A (#172) that has a role in Epstein-Barr virus replication and is the target of an antiviral drug, cordycepin that increases BCL7A methylation (turns gene expression off).

How can we leverage these biological insights? First, we inferred that identified antiviral factors are involved in the pathogenesis of a diverse spectrum of viruses. By focusing on common mechanisms of infection shared by diverse viral strains, we can identify existing broad-spectrum antiviral drugs that could be effective against SARS-CoV-2. For example, remdesivir, that interferes with viral replication machinery and was originally developed to treat Ebola, was subsequently found effective against SARS and MERS [49, 50] and may be effective against SARS-CoV-2 [51]. EIDD, another antiviral with a similar mechanism of action as remdesivir, has also been found to inhibit SARS-CoV-2 [52]. Some of the virus-associated genes are known or potential therapeutic targets, i.e., ITPR2 (#79), that can be inhibited by caffeine [53] and was recently identified as a phytochemical that could possibly remediate COVID-19 [54]. Because the virulence of SARS coronaviruses appears to be linked to selective inhibition of genes associated with antiviral IFN signaling [55, 56] and recent reports indicated that interferons improve recovery of COVID-19 patients [57], NIH has launched a randomized, controlled clinical trial of the combination of remdesivir and interferon beta-1a [58]. We also identified an additional explanation for the recent clinical findings showing the efficacy of dexamethasone in treating COVID-19 patients [59]; besides the likely prevention of inflammatory cardiac damage, our analysis suggests a role for dexamethasone-induced expression of cardioprotective genes such as ERBB4 (#90).

Second, we identified multiple genes that could be implicated in COVID-19 induced chronic symptoms such as myocarditis. For example, miR-1290-induced suppression of genes such as CACNB4 (#199), that normally promote expression of interferon-related genes in cardiac muscle cells, could worsen the cardiac effects of SARS-CoV-2, including myocarditis [60]. Notably, the predicted suppression of two genes linked to loss of smell, TENM1 and OLFM1, merits further investigation in COVID-19 survivors with lingering, chronic symptoms [26].

Third, these data show potential molecular mechanisms of differential COVID-19 susceptibility. A previous study showed that small variations in multiple genes can modify the penetrance of three diseases- familial hypercholesterolemia, hereditary breast and ovarian cancer and Lynch syndrome [61]- that have tier 1 evidence for interventions that reduce morbidity and mortality [62]. In the SARS-CoV-2-infected population, individual variations/polymorphisms or mutations in expression of pro-viral and antiviral genes could contribute to differences in disease severity and functional outcomes.

Fourth, the suggestion that variants or dysregulated expression of coronavirus-associated genes are risk factors for COVID-19 complications could inform treatment. For instance, single nucleotide polymorphisms (SNPs) in TSPYL4 (#285) are associated with pulmonary function and susceptibility to chronic obstructive pulmonary disease; thus, COVID-19 patients with this gene variant may experience more severe pulmonary symptoms. Because men are more likely to die of COVID-19 and women develop a more robust T cell immune response against SARS-CoV-2 than men [46], we can identify additional gender-specific risk factors by measuring expression levels of the genes with antiviral activity in all COVID-19 patients and determine if there is a more potent mobilization of genes and their associated cell signaling pathways in women.

One clear conclusion from this analysis is that this novel viral pathogen can potentially exploit hundreds of host factors to evade immune surveillance, survive and reproduce. Indeed, a recent study by Finkel et al., [44] showed that SARS-CoV-2 employs a multipronged strategy to coopt the host translation machinery and to suppress host defenses. Thus, drugs directed against single viral mechanisms are unlikely to be effective. We speculate that a more effective approach would involve a combinatorial therapeutic strategy against at least three or more host mechanisms, such as viral replication, inhibition of cell transport of viral particles, and enhancement of interferon-stimulated genes. Disease prognosis could also be improved by blood biomarker panels that identify genes and miRNAs that are altered in individual infected hosts. Studies of drugs in development shows that 73% of projects that are based on genetic links between target and disease are successful in Phase II trials, compared with the 43% success of projects not based on genetic links.

Finally, great therapeutic value could be derived from identifying endogenous host factors that prevent disease in the asymptomatic, COVID-19-positive population or increase risk of other diseases. Recognition that HIV-infected individuals with a mutation in the CCR5 gene were asymptomatic led to current drug therapies that allow a normal life in the HIV population [63]. Similarly, the observation that asymptomatic COVID-19 patients carry just as much virus in their nose, throat and lungs as those with symptoms and for as long as patients with symptoms [4], suggests an urgent need to identify protective host response factors that act as a natural defense against SARS-CoV-2. As we have shown here, recent reviews support the value of identifying the protective and pathogenic responses common to different coronaviral strains [64]. Notably, our study has identified the possible molecular mechanisms of COVID-induced loss of smell as well as genes whose dysregulation could lead to increased risk of developing diseases such as diabetes and cardiovascular disease and the chronic COVID-related health problems reported in survivors worldwide.

## Supporting information

**S1 Table. Predicted gene targets of miRNAs that are dysregulated by coronavirus infection and their pro-viral/antiviral functions.**
(PDF)

## Acknowledgments

The authors thank Andrew W. Hall and Clemmie White-Mathews for editorial support, and Christy Perry for preparing the illustrations and S1 Table.

## Author Contributions

**Conceptualization:** Stacy L. Sell, Helen L. Hellmich.

**Data curation:** Stacy L. Sell, Donald S. Prough, Harris A. Weisz, Steve G. Widen, Helen L. Hellmich.

**Formal analysis:** Harris A. Weisz, Steve G. Widen, Helen L. Hellmich.

**Investigation:** Helen L. Hellmich.

**Methodology:** Harris A. Weisz, Helen L. Hellmich.

**Project administration:** Helen L. Hellmich.

**Resources:** Donald S. Prough, Steve G. Widen, Helen L. Hellmich.

**Software:** Donald S. Prough, Helen L. Hellmich.

**Supervision:** Helen L. Hellmich.

**Validation:** Helen L. Hellmich.

**Visualization:** Helen L. Hellmich.

**Writing – original draft:** Stacy L. Sell, Helen L. Hellmich.

**Writing – review & editing:** Stacy L. Sell, Donald S. Prough, Helen L. Hellmich.

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
