## [Decision Letter · Decision Letter 0]

25 May 2021

PONE-D-21-09155

Leveraging publicly available coronavirus data to identify new therapeutic targets for COVID-19

PLOS ONE

Dear Dr. Hellmich,

Thank you for submitting your manuscript to PLOS ONE. After careful consideration, we feel that it has merit but does not fully meet PLOS ONE’s publication criteria as it currently stands. Therefore, we invite you to submit a revised version of the manuscript that addresses the points raised during the review process.

Please make the necessary changes as indicated by the reviewer.

We look forward to receiving your revised manuscript.

Kind regards,

Prasenjit Mitra, MD, MRSB, MIScT, FLS, FACSc, FAACC

Academic Editor

PLOS ONE

Journal Requirements:

Reviewers' comments:

Reviewer's Responses to Questions

**Comments to the Author**

1. Is the manuscript technically sound, and do the data support the conclusions?

Reviewer #1: Yes

2. Has the statistical analysis been performed appropriately and rigorously? 

Reviewer #1: Yes

3. Have the authors made all data underlying the findings in their manuscript fully available?

Reviewer #1: Yes

4. Is the manuscript presented in an intelligible fashion and written in standard English?

Reviewer #1: Yes

5. Review Comments to the Author

Reviewer #1: The authors have undertaken bioinformatic analysis of four coronavirus datasets to assess the differential expression of various anti-viral and pro-viral genes. The results have been discussed appropriately taking into consideration the numerous gene obtained from the analysis.

The following points may be addressed:

1. A short clinico-pathological comparative description of MERS to COVID & SARS CoV 1 to SARS CoV 2 indicating the probable relevance of the genes discussed would be appropriate.

2. Are there any original studies which have explored the temporal suppression of antiviral immune response in COVID when compared with other viral infections ? This may substantiate the temporal suppression of antiviral immune response observed in the analysis.

3. Line 254: How miR-139-3p expressed at low levels in both SARS-CoV-1 and SARS-CoV-2 cells as well as untreated control cells can be attributed to provide permissive environment ?

4. Line 364: It is unclear how the present analysis identify genes involved in COVID-19 induced chronic symptoms, rather it will give insight into COVID associated complication.

6. PLOS authors have the option to publish the peer review history of their article (what does this mean?). If published, this will include your full peer review and any attached files.

Reviewer #1: No

---

## [Author Response · Author response to Decision Letter 0]

8 Jul 2021

Reviewer #1: The authors have undertaken bioinformatic analysis of four coronavirus datasets to assess the differential expression of various anti-viral and pro-viral genes. The results have been discussed appropriately taking into consideration the numerous gene obtained from the analysis. 

The following points may be addressed: 

1.A short clinico-pathological comparative description of MERS to COVID & SARS CoV 1 toSARS CoV 2 indicating the probable relevance of the genes discussed would be appropriate.

Response: Respectfully, we would like to point out that the original intent of this in silico analysis was to identify biologically relevant host genes affected by the viral pathogens from previously deposited coronavirus datasets that may prove to be novel therapeutic targets and infer from published reports their functional relevance to host defenses. Our literature analysis showed that many of these pro-viral or antiviral host response genes were also targeted by many diverse viral pathogens known to infect human hosts. We did not intend to describe a comprehensive comparison of MERS, SARS-CoV-1 and SARS-COV-2 since hundreds of such comparisons have been published in the past year and a half since the beginning of the Covid-19 pandemic; our focus was on identifying novel gene targets of these viruses. To our knowledge, our in silico analysis of the gene targets of coronavirus-induced host genes is among the first to show that these host response genes are also implicated in the pathogenesis of many other diverse viruses; the implication is that broad spectrum antiviral drugs that are efficacious for other viral pathogens may be effective for Covid-19 patients but this remains to be proven in future studies. We also believe that the functional relevance of the genes have been addressed throughout the manuscript and further discussion would veer upon more speculation than necessary.

2.Are there any original studies which have explored the temporal suppression of antiviralimmune response in COVID when compared with other viral infections ? This may substantiatethe temporal suppression of antiviral immune response observed in the analysis.

Response: During the time we did the in silico analysis of these coronavirus datasets and wrote the manuscript, there were some unpublished reports in preprint servers suggesting that SARS-CoV-2 did indeed suppress the antiviral immune response as our analysis indicated. Most did not address the temporal suppression as described in our manuscript. Only recently were such reports published in peer-reviewed journals. We now cite, in the manuscript, one such supporting study by Finkel et al., that was recently published in Nature (https://www.nature.com/articles/s41586-021-03610-3).

3.Line 254: How miR-139-3p expressed at low levels in both SARS-CoV-1 and SARS-CoV-2cells as well as untreated control cells can be attributed to provide permissive environment ?

Response: In other words, since miRNA expression is inversely related to their gene targets, i.e. if high, their target genes are suppressed and if low, like miR-139-3p, their gene targets are highly expressed or upregulated, the naturally occurring expression of some host genes that are the targets of these miRNAs may help pathogenic viruses infect host cells. Since that was a

speculative comment on our part, we clarified this statement in the manuscript. What we meant was “ viruses could exploit and coopt the endogenously expressed host proteins to facilitate entry into host cells and favor the replication of viral proteins”.

4.Line 364: It is unclear how the present analysis identify genes involved in COVID-19 inducedchronic symptoms, rather it will give insight into COVID associated complication.

Response: The chronic symptoms described in the manuscript have been extensively documented in many published studies of hundreds of Covid-19 patients this past year and some of those studies are cited in our manuscript. We believe, based on the literature analysis of the function of the coronavirus-induced host gene targets, that not only does our study provide insight into Covid associated complications but also insight into the chronic symptoms reported by so called “Covid long haul” patients. For example, loss of smell is a frequently reported chronic health problem in Covid-19 survivors and the persistent symptoms of myocarditis, diabetes and so-called brain fog in Covid-19 survivors are among the chronic symptoms that can be linked to the dysregulated expression of some of the coronavirus gene targets mentioned in our study. We would like to point out that this study was not meant to be a comprehensive treatise of Covid-19-associated gene changes in human hosts; our intent was to identify novel gene targets that could be therapeutically targeted with existing or new coronavirus drugs in development. The four coronavirus datasets analyzed in this study represent the three recent and current causes of severe coronavirus outbreaks in human populations and were chosen without any other preconceived bias; we only present examples from the literature supporting the likely functional roles of the viral gene targets and we will leave it up to the scientific readership to infer the association of these genes with reported Covid symptoms based on the analysis presented in our study.

---

## [Decision Letter · Decision Letter 1]

15 Sep 2021

Leveraging publicly available coronavirus data to identify new therapeutic targets for COVID-19

PONE-D-21-09155R1

Dear Dr. Hellmich,

We’re pleased to inform you that your manuscript has been judged scientifically suitable for publication and will be formally accepted for publication once it meets all outstanding technical requirements.

Kind regards,

Prasenjit Mitra, MD, CBiol, MRSB, MIScT, FLS, FACSc, FAACC

Academic Editor

PLOS ONE

Additional Editor Comments (optional):

Reviewers' comments:

Reviewer's Responses to Questions

**Comments to the Author**

1. If the authors have adequately addressed your comments raised in a previous round of review and you feel that this manuscript is now acceptable for publication, you may indicate that here to bypass the “Comments to the Author” section, enter your conflict of interest statement in the “Confidential to Editor” section, and submit your "Accept" recommendation.

Reviewer #1: All comments have been addressed

2. Is the manuscript technically sound, and do the data support the conclusions?

Reviewer #1: Yes

3. Has the statistical analysis been performed appropriately and rigorously? 

Reviewer #1: Yes

4. Have the authors made all data underlying the findings in their manuscript fully available?

Reviewer #1: Yes

5. Is the manuscript presented in an intelligible fashion and written in standard English?

Reviewer #1: Yes

6. Review Comments to the Author

Reviewer #1: (No Response)

7. PLOS authors have the option to publish the peer review history of their article (what does this mean?). If published, this will include your full peer review and any attached files.

Reviewer #1: No

---

## [Editor Report · Acceptance letter]

20 Sep 2021

PONE-D-21-09155R1 

Leveraging publicly available coronavirus data to identify new therapeutic targets for COVID-19 

Dear Dr. Hellmich:

I'm pleased to inform you that your manuscript has been deemed suitable for publication in PLOS ONE. Congratulations! Your manuscript is now with our production department. 

Kind regards, 

on behalf of

Dr. Prasenjit Mitra 

Academic Editor

PLOS ONE